# Meta-Analysis of Transcriptomes in Insects Showing Density-Dependent Polyphenism

**DOI:** 10.3390/insects13100864

**Published:** 2022-09-23

**Authors:** Kouhei Toga, Kakeru Yokoi, Hidemasa Bono

**Affiliations:** 1Laboratory of BioDX, PtBio Co-Creation Research Center, Genome Editing Innovation Center, Hiroshima University, 3-10-23 Kagamiyama, Higashi-Hiroshima City 739-0046, Japan; 2Laboratory of Genome Informatics, Graduate School of Integrated Sciences for Life, Hiroshima University, 3-10-23 Kagamiyama, Higashi-Hiroshima City 739-0046, Japan; 3Insect Design Technology Module, Division of Insect Advanced Technology, Institute of Agrobiological Sciences, National Agriculture and Food Research Organization (NARO), 1-2 Owashi, Tsukuba 305-8634, Japan

**Keywords:** meta-analysis, density-dependent polyphenism, aphid, locust, RNA sequencing, gene set enrichment analysis

## Abstract

**Simple Summary:**

Population density can be an environmental cue to induce modification of insect morphology, physiology, and behavior. This phenomenon is called density-dependent plasticity. Aphids and locusts exhibit textbook examples of density-dependent plasticity but there is a lack of integrative understanding for insect density-dependent plasticity. To address this problem, we combined public gene expression data obtained from multiple studies and re-analyzed them (this process is called meta-analysis). The present study provides additional insight into the regulatory mechanisms of density-dependent plasticity, demonstrating the effectiveness of meta-analyses of public transcriptomes.

**Abstract:**

With increasing public data, a statistical analysis approach called meta-analysis, which combines transcriptome results obtained from multiple studies, has succeeded in providing novel insights into targeted biological processes. Locusts and aphids are representative of insect groups that exhibit density-dependent plasticity. Although the physiological mechanisms underlying density-dependent polyphenism have been identified in aphids and locusts, the underlying molecular mechanisms remain largely unknown. In this study, we performed a meta-analysis of public transcriptomes to gain additional insights into the molecular underpinning of density-dependent plasticity. We collected RNA sequencing data of aphids and locusts from public databases and detected differentially expressed genes (DEGs) between crowded and isolated conditions. Gene set enrichment analysis was performed to reveal the characteristics of the DEGs. DNA replication (GO:0006260), DNA metabolic processes (GO:0006259), and mitotic cell cycle (GO:0000278) were enriched in response to crowded conditions. To date, these processes have scarcely been the focus of research. The importance of the oxidative stress response and neurological system modifications under isolated conditions has been highlighted. These biological processes, clarified by meta-analysis, are thought to play key roles in the regulation of density-dependent plasticity.

## 1. Introduction

Phenotypic plasticity is the ability to respond adaptively to environmental variations by producing more than one phenotype for a single genotype [1,2]. If two or more distinct phenotypes are produced, this is called polyphenism [3]. Various environmental conditions can be cues to induce plastic phenotype changes, with population density being one of those [3,4,5,6]. Locusts and aphids are representative insect groups for which density-dependent plasticity is considered to have contributed to their evolutionary success [7,8]. Aphids exhibit wing polyphenotypes and locusts exhibit polyphenisms in body color and behavior (e.g., gregarious and flight ability).

Physiological mechanisms underlying density-dependent polyphenism have been revealed in aphids [7] and locusts [9]. Furthermore, a recent transcriptome analysis revealed gene expression profiles in response to different density conditions. RNA sequencing (RNA-seq) revealed that a laterally transferred viral gene is responsible for switching wing polyphonic phenotypes [10]. In locusts, expressed sequence tag (EST), microarray, and RNA-seq revealed differentially expressed genes (DEGs), including juvenile hormone (JH)-binding protein [11], heat-shock proteins [12], oxidative stress response genes [12], neurotransmitter-related genes [13], catecholamine metabolic pathway genes [14], and the microtubule cytoskeleton [15]. However, the common molecules involved in all locust species exhibiting density-dependent polyphenism remain largely unknown. As mentioned above, RNA sequencing has revealed the species-specific regulatory mechanisms in each species, and the common genes involved in the regulation of density-dependent polyphenism are predicted to be present. However, an inclusive analysis of aphids and locusts was not performed to determine the common genes expressed in all or multiple species that show density-dependent polyphenism.

Meta-analysis that combines the transcriptome results from multiple studies is thought to be effective in providing additional insights into density-dependent polyphenisms because a meta-analysis can uncover new information that cannot be achieved with conventional hypothesis-driven research methods. Meta-analysis of transcriptomes is effective in revealing novel gene expression profiles [16,17,18,19,20]. As mentioned above, the amount of transcriptome data related to density-dependent polyphenism is sufficient to execute the meta-analysis. In the present study, we conducted a meta-analysis of public transcriptomes using RNA-seq data obtained from five *Schistocerca* species, the pea aphid *Acyrthosiphon pisum*, and the migratory locust *Locusta migratoria*. We identified DEGs between crowded and isolated conditions and performed gene set enrichment analysis to understand the characteristics of the DEGs.

## 2. Materials and Methods

### 2.1. Curation of Public Gene Expression Data

Expression data were searched for in the public databases Gene Expression Omnibus (GEO) [21] and DBCLS SRA (https://sra.dbcls.jp/ (accessed on 16 April 2021)), and RNA-seq data archived in the Sequence Read Archive (SRA) [22] using the keywords including crowding, density, polyphenism, and gregarious. The datasets examined in this study were selected based on the following criteria: RNA-seq data obtained from crowded or isolated insect species showing density-dependent polyphenism.

### 2.2. RNA-Seq Data Retrieval, Processing, and Quantification

SRA retrieval and conversion to FASTQ files was performed using the prefetch and fasterq-dump programs in the SRA Toolkit (v2.9.6) [23], respectively. To decrease disk space usage, the obtained FASTQ files were immediately compressed using pigz (v. 2.4). For the trimming and quality control, Trim Galore! (v.0.6.6) [24] and Cutadapt (v.3.4) [25] software was used. Trim Galore! [24] was run with the parameters -fastqc --trim1 -paired (if data were paired-end). Trim Galore! [24] includes Cutadapt (v.3.4) [25] to trim low-quality base calls. Since all data were passed through this quality check, low-quality data were not included in this study.

### 2.3. Quantification of Gene Expression Level

In species whose genomic information is available (the pea aphid *Acyrthosiphon pisum* and the migratory locust *Locusta migratoria*), RNA-seq reads were mapped to reference genome sequences using HISAT2 (v.2.2.1) [26] with parameters -q --dta -x (*Acyrthosiphon pisum*: GCF_005508785.1 or *L. migratoria*: GCA_000516895.1). Quantification of the mapped data was performed using StringTie (2.1.5) [27], and the values of transcripts per million (TPM) were used as quantitative values of gene expression.

In species with genomic information not available (all species of grasshopper *Schistocerca*), TPM values were calculated using Salmon (v.1.5.0) [28] with parameters -i index -l IU. Transcriptome assemblies required for Salmon were retrieved from the Transcriptome Shotgun Assembly (TSA) database (Appendix A). If there were no available assemblies in TSA, de novo transcriptome assembly was performed using Trinity (v2.13.1) program with Docker environment utilizing scripts in the Systematic Analysis for Quantification of Everything (SAQE) repository [16]. Trinity was performed with the default parameters.

### 2.4. The Detection of Differentially Expressed Genes

To evaluate the changes in expression of each gene, the expression ratio was calculated using TPM paired with crowded and isolated transcriptomes (CI ratio). The CI ratio for each gene was calculated using Equation (1):(1)$$\mathrm{CI}\;\mathrm{ratio}=\log_{2}\left({\mathrm{TPM}}_{\mathrm{crowded}}+0.01\right)-\log_{2}\left({\mathrm{TPM}}_{\mathrm{isolated}}+0.01\right),$$

This calculation was performed for all pairs of crowded and isolated transcriptomes. Pairs of transcriptome data used for comparison are shown in Appendix A. The value “0.01” was added to the TPM for each gene to convert zero into a logarithm. Crowded and isolated ratios (CI-ratio) were classified as “upregulated”, “downregulated”, or “unchanged” according to the thresholds. Genes with a CI ratio greater than 1 (2-fold expression change) were treated as “upregulated”. Genes whose CI ratio was under –1 (0.5-fold expression changes) were treated as “downregulated”. Others were treated as “unchanged”. To reveal the DEGs between the crowded and isolated conditions, the crowded and isolated scores (CI score) of each gene were calculated by subtracting the number of sample pairs with “downregulated” from those of “upregulated”, as shown in Equation (2):(2)$$\mathrm{CI}\;\mathrm{score}={\mathrm{count}\;\mathrm{number}}_{\mathrm{upregulated}}-{\mathrm{count}\;\mathrm{number}}_{\mathrm{downregulated}},$$

The calculation method for CI ratios and CI scores was based on a previous study [17] (https://github.com/no85j/hypoxia_code/blob/master/CodingGene/HN-score.ipynb (accessed on 17 March 2022)). Visualization of CI ratio was performed with heatmap2 (R ver 4.2.0).

### 2.5. Gene Annotation and Gene Set Enrichment Analysis

In species whose genomic information is available, GFFread (v0.12.1) [29] was used with the reference genome (*Acyrthosiphon pisum*: GCF_005508785.1 and *L. migratoria*: GCA_000516895.1) to extract the transcript sequence from the genome data. In species whose genomic information is not available (all species of the *Schistocerca* genus), transcript sequences were collected from the TSA database (Appendix A). However, transcript sequences of *Schistocerca gregaria* were obtained by de novo assembly using Trinity, as described above (2.3. Quantification of gene expression level). TransDecoder (v5.5.0) (https://transdecoder.github.io/ accessed on 22 June 2021) was used to identify coding regions and translate them into amino acid sequences of all species. Open reading frames were extracted by TransDecoder.Longorfs with parameters -m 100, and coding regions were predicted by TransDecoder.Predict with default parameters. Then, the protein BLAST (BLASTP) program with parameters -evalue 0.1 -outfmt 6 -max_target_seqs 1 in the NCBI BLAST software package (v2.6.0) was used to determine the orthologous gene relationship among species. Ensembl Biomart was used to obtain stable protein IDs and gene names from *D. melanogaster* gene set (BDGP6.32).

Gene set enrichment analysis was performed by Metascape [30] using the DEGs obtained in Section 2.4.

## 3. Results

### 3.1. Data Collection of Transcriptomes in Density-Dependent Polyphenism

To compare gene expression under crowded and isolated conditions, we retrieved transcriptome data acquired under different density conditions. Consequently, 66 pairs of transcriptome data were collected from the public databases (Table 1). Full information about the dataset used in this study is provided in Appendix A [10,31,32,33,34,35]. These data comprised transcriptomes from seven insect species, including aphids and locusts. The tissues used to obtain these transcriptomes are shown in Table 1.

### 3.2. Gene Set Enrichment Analysis

After the expression ratio (CI ratio) was calculated for each species, as described in the Materials and Methods Section 2.4, we integrated the CI ratio table and obtained the CI ratio values of 2652 genes (Appendix A). During this process, orthologous relationships among species were required to be determined using BLAST, because the gene names of the expression ratio are species specific depending on the annotation file or transcriptome assemblies of each species. First, we performed a BLASTP search using all genes in *S. gregaria* as queries against those in each species examined in this study. Since density-dependent polyphenisms have been extensively studied using this species, this species was selected as a reference for the determination of orthologous relationships among species. Next, we performed searches for all genes vs. all BLASTP between *D. melanogaster* and *S. gregaria* to annotate the gene names of each species into those of *D. melanogaster*.

CI scores were calculated based on the CI ratio table (Appendix A). The previous studies showed that *ebony*, *Mcm2*, and *Mcm7* were upregulated in response to crowded condition, and *Duox*, *Mdr49* and *Cyp6a14* were downregulated in response to crowded condition (i.e., upregulated in response to isolated condition) [14,33,34]. In this study, the CI score of these genes showed high or low values (Figure 1). This means that the expression patterns of these genes were consistent with previous results. Previous transcriptome analyses also showed that GO:0008061 [cuticle binding] and GO:0006979 [oxidative stress] were enriched in crowded and isolated condition, respectively [14,33,34]. CI score of genes that was included in these GO terms (GO:0008061; *obst-E*, GO:0006979; *spz*, *Glaz*, *srl*) showed high or low values (Figure 1). These results show the suitability of the CI score method to detect the DEGs. We listed the top 1% of genes with the highest and lowest CI scores (Table 2 and Table 3, respectively). *ebony* (*e*) showed a high CI score for the upregulated DEGs (tenth rank in Table 2). This gene, encoding a protein that converts dopamine to N-β-alanyl dopamine, is highly expressed in gregarious *L. gregaria*, leading to the production of the yellow color [14]. In the top 1% of genes with the highest and lowest CI scores, we assessed the expression pattern in each species and/or tissue (Appendix A). Many genes with unchanged expression levels were observed in *A. pisum* and *L. migratoria*. Although this tendency was also observed in the heatmap of expression ratio (Appendix A), the genes that showed a drastic expression change were observed in *L. migratoria* samples (SmD3 and CG34461).

We selected 97 upregulated genes and 199 downregulated genes as DEGs (CI score thresholds:8 and −8). Using these DEGs, we performed gene set enrichment analysis using Metascape to identify their characteristics. Metascape showed that “DNA replication” (GO:0006260; *CG14803*, *CG2990*, *Psf3*, *CG8142*, *RnrL*, *Psf1*, *Pol31*, *Mcm7*, *Mcm2*, *E2f1*, *DNApol-alpha60*, *dup*), “DNA metabolic process” (GO:0006259; *CG14803*, *CG2990*, *Psf3*, *CG8142*, *RnrL*, *FANCI*, *pds5*, *Psf1*, *Pol31*, *Gen*, *Mcm7*, *Mcm2*, *pch2*, *Fancl*, *CG5285*, *E2f1*, *DNApol-alpha60*, *dup*), and “mitotic cell cycle” (GO:0000278; *Psf3*, *sofe*, *borr*, *Incenp*, *pds5*, *SmD3*, *tum*, *Klp61F*, *Psf1*, *Pol31*, *eco*, *polo*, *Mcm2*, *E2f1*, *jar*, *Cap-D2*, *DNApol-alpha60*, *dup*, *Zwilch*) were enriched significantly in upregulated DEGs (Figure 2A). In addition, we searched for genes that were likely to be associated with density-dependent polyphenism (Table 2). JHEH encodes for juvenile hormone epoxide hydrolase involved in JH inactivation, as reported by Iga and Kataoka 2012 [36]. JH plays a central role in the regulation of polyphenism.

The response to oxygen-containing compounds (GO:1901700; *G9a*, *hfw*, *Shmt*, *Clic*, *Duox*, *FASN1*, *Kr-h1*, *GLaz*, *ple*, *Rab8*, *srl*, *Syn*, *CASK*, *spz*, and *dco*) was significantly enriched in the downregulated DEGs (Figure 2B). Multiple genes involved in the response to oxidative stress (GO:0006979: *Duox*, *GLaz*, *ple*, *srl*, *and spz*) were included in GO: 1,901,700.

The CI score of the *pale* (*ple*) gene, encoding tyrosine hydroxylase, which is involved in pigmentation of the cuticle and catecholamine biosynthesis [37], was the second lowest (Table 3). This indicates that the expression of *ple* was higher in many isolated transcriptomes than in crowded transcriptomes. *ple* genes are known to be highly expressed under crowded conditions, inducing gregarious color and behavior [14].

Among the downregulated DEGs, the genes that are reported to function in the nervous system were conspicuous (*CG9657*: [38], *Lrp4*: [39], *GluClalpha*: [40], *Syt4*: [41]).

## 4. Discussion

As various data acquired under different conditions are registered, manual curations are needed for meta-analysis. In this study, we collected RNA sequencing data (66 pairs) from public databases and compared their expression levels in various insect species. We aimed to identify the common genes that can explain the regulation of the density-dependent polyphenism of all or multiple species. However, in the gene list with highest or lowest CI score (Table 2, Table 3, Appendix A), almost all genes showed no expression changes in *A. pisum.* This result may reflect the use of different developmental stages. The aphids used in this study are derived from adults, whereas the other samples are derived from larval stages. However, we believe that our meta-analysis of public RNA sequencing using this vast amount of data has led to important insights into the gene expression profile underlying density-dependent polyphenism, as discussed below.

Gene set enrichment analysis showed that DNA replication, DNA metabolic processes, and the mitotic cell cycle were enriched in response to crowded conditions (Figure 1). DNA replication and cell cycle have rarely been focused on as regulatory mechanisms of density-dependent polyphenism [33]. A recent study [42] showed that the regulation of DNA replication and the cell cycle is involved in the density-dependent wing polyphenism of the planthopper *Nilaparvata lugenes*. Together with this study, our results emphasize the importance of DNA replication and the cell cycle as the regulatory mechanisms of density-dependent polyphenism. Although wing polyphenism was also observed in *A. pisum*, it was not observed in locusts. Therefore, the regulation of DNA replication and cell cycle is expected to be involved in other developmental processes in locusts. Although the role of JH in aphid wing polyphenism is controversial, JH content is lower under crowded conditions than under isolated conditions in locusts [9,43]. Therefore, Jheh2 expression in response to crowded conditions may play an important role in JH degradation.

The high expression of *ple* gene under isolated conditions was not consistent with a previous study. Although *ple* genes are known to be responsible for gregarious pigmentation in *L. migratoria*, the relationship between *ple* genes and body-color polyphenism is controversial [44]. Several downregulated DEGs (*Duox*, *GLaz*, *ple*, *srl*, *and spz*) classified under response to oxygen-containing compounds were related to the response to oxidative stress, and *ple* was included in that category. A previous microarray study showed that the expression of transcripts encoding proteins against oxidative stress damage (peroxiredoxin, 5-oxoprolinase, microsomal glutathione-S-transferase, and transaldolase) was higher in isolated locusts than in crowded locusts [12]. Rapid accumulation of oxidative stress inhibits flight sustainability in solitary *L. migratoria*, leading to variations in flight traits between solitary and gregarious locusts [45]. Therefore, the high *ple* expression under the isolated conditions in this study may result from the response to oxidative stress.

Neurological system modifications may play an important role in inducing density-dependent phenotypic changes in *S. gregaria* [12] and *L. migratoria* [15]. We found that the expression of several genes functioning in the nervous system was increased under isolated conditions (CG9657 [38], Lrp4 [39], GluClalpha [40], and Syt4 [41]). CG9657 is an SLC5A transporter expressed in glial cells and is involved in sleep behavior in *D. melanogaster* [38]. Presynaptic *Lrp4* functions to ensure normal synapse numbers in *D. melanogaster* [39]. *GluClalpha* plays an important role in the ON/OFF selectivity of the visual systems in *D. melanogaster* [40]. Syts are Ca^2+^-binding proteins involved in the presynaptic transmitter release [41]. These genes are also related to density-dependent behavioral changes.

## 5. Conclusions

We identified novel genes related to density-dependent polyphenism in insects using a meta-analysis of public transcriptomes. Reliable and general principles should be derived from meta-analysis because this method integrates the results of a number of studies. Therefore, our results can be applied to other species that exhibit density-dependent polyphenisms.

## Figures and Tables

**Figure 1 insects-13-00864-f001:**
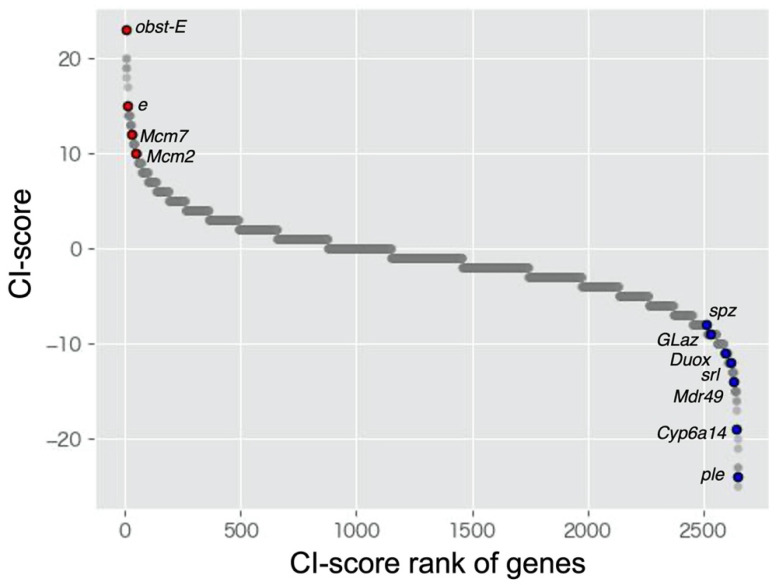
Scatter plots of CI-score of all genes (2652 genes) identified in this study. The red and blue dots indicate genes that have been identified in the previous studies as the differentially expressed genes between crowded and isolated conditions.

**Figure 2 insects-13-00864-f002:**
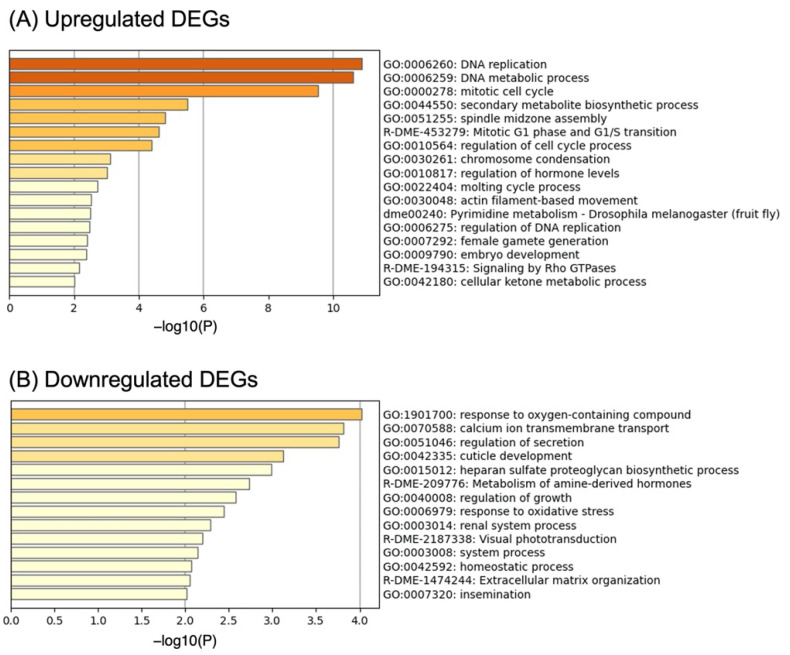
Results of gene set enrichment analysis in (**A**) upregulated DEGs and (**B**) downregulated DEGs.

**Table 1 insects-13-00864-t001:** Summary of datasets used in this study.

Species	Number of RNA Sequencing Files (under Crowded Condition)	Number of RNA Sequencing Files (under Isolated Condition)	Tissue
*Schistocerca gregaria*	1	1	CNS
*Acyrthosiphon pisum*	8	8	whole
*Schistocerca americana*	10	10	head, thorax
*Schistocerca nitens*	10	10	head, thorax
*Schistocerca piceifrons*	10	10	head, thorax
*Schistocerca serialis cubense*	10	10	head, thorax
*Locusta migratoria*	17	17	brain, Integument, thoracic ganglion

**Table 2 insects-13-00864-t002:** Top 1% of genes with the highest CI scores.

	Number of Samples with Expression Patterns Responding to the Crowded Condition			
Gene_Name	Upregulated	Downregulated	Unchanged	CI Score
obst-E	31	8	27	23
CG14803	23	3	40	20
His2A:CG33835	30	10	26	20
Jheh2	22	3	41	19
nw	27	8	31	19
HDAC1	21	3	42	18
Ugt36D1	24	7	35	17
CG5321	22	7	37	15
Cdep	21	6	39	15
e	17	2	47	15
pds5	19	4	43	15
CG4572	21	7	38	14
CG6765	16	2	48	14
Incenp	23	9	34	14
Psf3	21	7	38	14
ft	21	7	38	14
polo	19	5	42	14
tou	22	8	36	14
tum	19	5	42	14
CG10175	22	9	35	13
CG2990	20	7	39	13
CG8173	18	5	43	13
DNApol-alpha60	22	9	35	13
SmD3	18	5	43	13
Ts	20	7	39	13
CG8646	20	8	38	12

**Table 3 insects-13-00864-t003:** Top 1% of genes with the lowest CI scores.

	Number of Samples with Expression Patterns Responding to the Crowded Condition			
Gene_Name	Upregulated	Downregulated	Unchanged	CI Score
CG14301	3	28	35	−25
ple	7	31	28	−24
CG9657	6	29	31	−23
Lrp4	3	26	37	−23
GluClalpha	5	26	35	−21
CG34461	9	29	28	−20
Cyp6a14	9	28	29	−19
CG10211	10	27	29	−17
Ggt-1	8	24	34	−16
Syt4	6	22	38	−16
Cralbp	9	24	33	−15
KaiR1D	3	18	45	−15
RpS5a	3	18	45	−15
Shmt	6	21	39	−15
Tcs3	4	19	43	−15
tyn	2	17	47	−15
CG13744	7	22	37	−15
CG15449	5	19	42	−14
Est-6	9	23	34	−14
KaiR1D	11	25	30	−14
Mdr49	7	21	38	−14
dpr12	6	20	40	−14
CG13643	7	20	39	−13
CG6006	5	18	43	−13
CG7246	2	15	49	−13
Cnb	5	18	43	−13

## Data Availability

The data presented in this study are openly available in figshare [https://doi.org/10.6084/m9.figshare.20174255.v1 (accessed on 30 June 2022)].

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
