# Peer review of "Meta-Analysis of Transcriptomes in Insects Showing Density-Dependent Polyphenism"

_insects, 2022, doi:10.3390/insects13100864_

Round 1

Reviewer 1 Report (Previous Reviewer 1)

Authors Toga et al. have revised their manuscript implementing some of the reviewer's comments. Unfortunately the most important concerns about the data integration/quality and the statistical methods still hold. E.g.: fold-change-like metrics (the authors' CI score) are not sufficient to select robust differential genes in NGS data, and read trimming can not save low-quality samples/datasets. Without careful quality-based data selection and without sound statistical methods, it is not possible to recommend the publication of this work.

Author Response

Reviewer 2 Report (Previous Reviewer 2)

Drs. Toga, Bono, and now Yokoi have done extensive modifications to the manuscript as requested. This reviewer appreciates the further explanations as well as the increased data availability in the supplementary figures and tables. 

Author Response

Reviewer 3 Report (New Reviewer)

In this study, a meta-analysis of public transcriptomes was performed to gain additional insights into the molecular underpinning of density-dependent plasticity. Results conclude that DNA replication (GO:0006260), DNA metabolic processes (GO:0006259), and mitotic cell cycle (GO:0000278) were enriched in response to crowded conditions. The importance of the oxidative stress response and neurological system modifications under isolated conditions has been highlighted. I think this is an excellent work, and very interesting. The results of this study can be applied to other species that exhibit density-dependent polyphenisms. In addition, the manuscript was well written and the methodology used in this study is correct. I suggest this work could be considered to be published in Insects journal.

This manuscript is a resubmission of an earlier submission. The following is a list of the peer review reports and author responses from that submission.

Round 1

Reviewer 1 Report

In the manuscript, authors Toga and Bono present a meta-analysis of RNA-seq data related to insects in order to discover genes associated with density-dependent polyphenism.

Althought such an analysis could be of interest to the community, the analysis is lacking important information to make it reproducible, and usage of powerful statistics. It is also quite limited in scope as the authors have performed quite a limited number of analyses. Moreover, the data integration is also lacking a strict quality control procedure to ensure that low-quality data is not included in the analysis. Such low-quality data is unfortunately very common in the public databases from which the authors take the data. 

Major concerns:

8) The analysis misses a detailed quality control analysis of each dataset and each file in order to decide if the data of various sources can be analysed together

1) usage of statistics to derive differential genes is quite poor (CI ratio and CI score). Authors should use more powerful methods to define the genes of interest (e.g. specific statistics for meta-analyses, or test-based methods).

2) Several bioinformatics tools are used to process the data but the authors need to detail the parameters and values used for each tool. 

3) Dataset identifiers must be also provided for good reporting

4) CI ratio: why do authors use log10 and not log2 to transform TPM values? The latter would be a more standard approach.

5) Table S3 and S4 were not available to the reviewer

6) S. gregaria is used as a reference: why does it have only 1 pair of files for the analysis (contrary to other species with several pairs)? The relevance of the differential genes derived for the reference is very low as it is from only 2 files.

7) After deriving differential genes, the authors perform only a standard gene-set enrichment analysis before concluding the paper. As the results to the gene set enrichment analysis is not very specific, it limits quite drastically the scope and interest of this work. 

Minor points:

1) Section 2.5 needs English editing. At this point, it is unclear what method is used for which species. 

2) Should Line 136 refer to section 2.4 instead of 2.3?

Reviewer 2 Report

Drs. Toga and Bono have produced a meta-analysis of transcriptomes from insects exhibiting environmentally-induced plasticity depending on population density. By combining publicly available datasets of seven different species who exhibit this form of plasticity, they were able to score genes based on whether they were expressed in crowded or isolated conditions. While the experimental design is interesting and more research into the molecular etiology of this plasticity is needed, the results need improvement with how they are presented.

Major points

In this analysis, several species of insects were used. Additionally, these datasets also contained different tissues. When making comparisons and identifying novel genes related to a specific phenotype, this can be problematic. Each cell type (and at a larger scale, each tissue) has its own transcriptome. With some samples, the whole insect was used, with others, the whole head or whole thorax. In L. migratoria, more specific organs were used. While limitations on the amount of publicly available datasets are understandable, why combine such vastly different tissue datasets? This can impact results for several different hormonal pathways as what is expressed in the brain or head, may not be in the thorax

More importantly, why do an inclusive analysis with the pea aphid and migratory locus in addition to the 5 species of Schistocerca? The pea aphid in particular only has 8 samples out of 66. Considering that the aphid is in a different order than the other included species (grasshopper and locust) and it is only a small fraction of the overall data, why was it included? Reference 16 had a similar analysis where D. melanogaster and C. elegans were compared to find overlapping genes, however there was no correlated expression between the two species and there was minimal overlap of differentially expressed genes. It is therefore confusing why the authors would try a similar inclusive analysis as opposed to including only similar species as this may prove detrimental to the results.

The stage of life in the tissues was also not discussed. What instar were the samples from? This could be another confounding factor especially as juvenile hormone is differentially expressed at different instar stages and one of the main results discussed was JHEH. Overall, the sample sources need to be discussed more and discussed as potential limitations to the study.

Tables 2 and 3 were very confusing and ultimately I did not feel they were very valuable to demonstrate patterns in gene expression. This may be better represented by a heat map, especially if this heat map labeled samples by species or genus. This representation may help my earlier points with the issues of different species (and order) in addition to different tissue

Minor points

In lines 66-68, the authors state that this form of meta-analysis is effective in revealing novel gene expression profiles, a point that this reviewer agrees on. However the cited references (16-18) are self-citations with Dr. Bono as the corresponding author for all. It may be beneficial to add outside studies when supporting this argument as opposed to strictly self-citation.

In line 98 and going forward in the paper, the word “salmon” should be capitalized to “Salmon.” This can momentarily confuse the reader into thinking you mapped the grasshopper genome to salmon genome as opposed to using the tool Salmon, especially if they are unfamiliar with the script.

Line 112 – is this referring to log2 fold change? That was assumed but not immediately obvious

Line 173 – Ebony (e) is mentioned without any explanation as to what it is. It is also stated that it has a high CI score for upregulated DEGs, but It is not in any of the tables. Please clarify.

Tables S3 and S4 were not included in the supplemental information to download with the paper. While I did find these tables thanks to the link in the text, they were not appropriately labeled as S3 or S4. It would be nice if they were all downloadable in one spot

It may be beneficial to expand on the names of several of the major genes mentioned. For example, stating along the lines that JHEH is juvenile hormone epoxide hydrolase, an enzyme that breaks down JH. Not all genes are necessary, but it may help readers who are not as familiar with each major gene having a sentence or two devoted to it in results or discussion.